# Exploring Context-Aware Evaluation Metrics for Machine Translation

**Xinyu Hu** and **Xunjian Yin** and **Xiaojun Wan**
Wangxuan Institute of Computer Technology, Peking University
{huxinyu,xjyin,wanxiaojun}@pku.edu.cn

## Abstract

Previous studies on machine translation evaluation mostly focused on the quality of individual sentences, while overlooking the important role of contextual information. Although WMT Metrics Shared Tasks have introduced context content into the human annotations of translation evaluation since 2019, the relevant metrics and methods still did not take advantage of the corresponding context. In this paper, we propose a context-aware machine translation evaluation metric called Cont-COMET, built upon the effective COMET framework. Our approach simultaneously considers the preceding and subsequent contexts of the sentence to be evaluated and trains our metric to be aligned with the setting during human annotation. We also introduce a content selection method to extract and utilize the most relevant information. The experiments and evaluation of Cont-COMET on the official test framework from WMT show improvements in both system-level and segment-level assessments.

## 1 Introduction

Automatic evaluation metrics play an important role in the field of machine translation (MT) and promote the advancement of MT research. Many traditional metrics, such as BLEU (Papineni et al., 2002) and METEOR (Banerjee and Lavie, 2005), were earlier proposed based on lexical-level matching between the human reference and MT-generated hypothesis. Despite excellent efficiency and ease of use, some recent work (Kocmi et al., 2021; Freitag et al., 2022) has proven that these string-based metrics like BLEU perform poorly in terms of consistency with human evaluation. Instead, many metric models (Lo, 2019; Mukherjee et al., 2020; Sellam et al., 2020; Rei et al., 2021; Perrella et al., 2022; Wan et al., 2022; Rei et al., 2022) using neural networks have emerged recently and achieved great capabilities for MT evaluation with the help of large-scale human annotations.

However, these studies focused primarily on the evaluation of the single sentence, while content of the document that contains it should also be considered. The sentence itself may be semantically ambiguous and thus require surrounding sentences for more proper understanding and evaluation. In fact, some work has improved document-level machine translation by introducing contextual information. Furthermore, the WMT Shared Task (Kocmi et al., 2022), which mainly contributes to the annotation data of MT evaluation, has incorporated contextual information into the process of human annotation since 2019. Unfortunately, most existing methods ignored the effect of context and only trained the metrics at the sentence level. Vernikos et al. (2022) proposed additionally encoding previous sentences during the calculation of evaluation scores. Although they achieved improvement, their attempts were not comprehensive enough and performed poorly on segment-level correlation.

In this paper, we further explore context-aware methods for MT evaluation and propose our trained metric, Cont-COMET. To align with the annotation setting of human evaluation, we improve and train our model with additional encoded contextual information based on the COMET (Rei et al., 2020) framework, which has shown promising performance. In particular, we simultaneously utilize both the preceding and subsequent nearby contexts and attempt different numbers of sentences. Furthermore, given that the document is usually too long for the model to encode entirely, we propose a context selection method to extract the most relevant portion of the document as additional context. We evaluate our metric on the official Multidimensional Quality Metrics (MQM) judgments from WMT21 (Freitag et al., 2021) and WMT22 (Freitag et al., 2022). The experimental results demonstrate our Cont-COMET achieves great improvements in segment-level assessment and better system-level correlation compared with previous work.

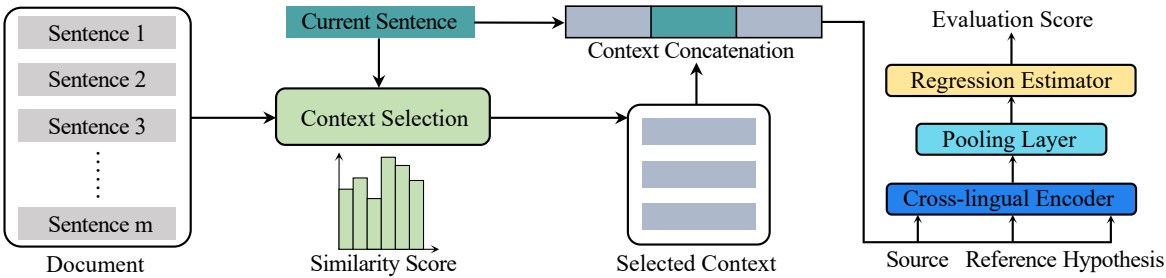

Figure 1: The brief framework of our context-aware training and context selection method for MT evaluation.

In general, our primary contributions are as follows: (1) We observe the gap in context utilization between current neural metrics and human annotation from WMT and improve the training approach with additional context. (2) We explore different incorporations of surrounding sentences and propose a context selection method to extract more important content. (3) We collect and collate the annotated data with document information and accordingly train our metric Cont-COMET, which achieves overall improvements.[1]

## 2  Our Methods

We first introduce the COMET framework, which was proposed for MT evaluation and achieved superior performance in the annual WMT Metrics Shared Task. The basic architecture consists of three main components: the cross-lingual encoder, layer pooling, and estimator module. To leverage the knowledge of pretrained language models, the XLM-RoBERTa-large (Conneau et al., 2020) is served as the cross-lingual encoder. It generates the representation $e^l = (e_0^l, e_1^l, \cdots, e_n^l)$ of the input sequence $x = (x_0, x_1, \cdots, x_n)$ for each layer $l$. Then these representations are fused into a single embedding sequence through a trained layer-wise attention mechanism. And the average pooling is employed to derive the final sentence embedding for each input. We let $s$, $r$, and $h$ refer to the final embeddings of source, reference, and MT-generated hypothesis, respectively. A feed-forward regression estimator is trained according to the combined vector $v = [h; r; h \odot s; h \odot r; |h - s|; |h - r|]$, where $\odot$ represents the element-wise product. More details about COMET can be found in Rei et al. (2020).

### 2.1  Context-aware Training

In summary, we improve the original training approach based on the COMET framework with ad-

ditional encoding of context content. In previous work, the representations of the source ($S$), reference ($R$), and hypothesis ($H$) for each sentence were directly encoded and average-pooled. Instead, we aim to incorporate contextual information when processing the sentence to be evaluated, so as to align with the human annotation setting. Unlike the work of Vernikos et al. (2022), we utilize the content of both the previous and next sentences adjacent to the current sentence, denoted as $S_{prev}, R_{prev}, H_{prev}$ and $S_{next}, R_{next}, H_{next}$.

We then conduct context-aware encoding, as shown in Figure 1. As for reference, the current $R_{curr}$ is concatenated with surrounding content in the original order, resulting in an expanded sequence $R = [R_{prev}; R_{curr}; R_{next}]$. The positions of the tokens in the current sentence are masked as $p_{curr}$. Then, we encode $R$ and apply average pooling to the embeddings at the position $p_{curr}$ to obtain the context-aware representation:

$$r_c = \text{Average\_pooling}(\text{Encoder}(R), p_{curr})$$

And the corresponding combined vector for the regression estimator becomes $v_c = [h_c; r_c; h_c \odot s_c; h_c \odot r_c; |h_c - s_c|; |h_c - r_c|]$.

The expanded content of the source and reference consists of their respective context sentences, while the hypothesis is concatenated with the conext of reference, namely $H = [R_{prev}; H_{curr}; R_{next}]$. It can avoid error propagation in the hypothesis and was also employed by Vernikos et al. (2022). And our following experiments prove that this setting is also superior for our trained metric. In addition, we conduct ablation studies to explore the effects of different usages of context sentences and training approaches.

### 2.2  Context Selection

Despite the intuition that using as much context as possible should be beneficial to the evaluation performance, the maximum input length of the XLM-

[1]Our context-aware dataset and Cont-COMET metric are available at https://github.com/herrxy/Cont-COMET.

RoBERTa-large used in COMET is limited to 512 tokens. In many cases, the entire document actually exceeds the length, and in the meantime, the effect of encoding may drop as the processed input becomes longer. Therefore, instead of directly concatenating adjacent context sentences, we propose a context selection method to extract more relevant context for the evaluation. We aim to take better advantage of the capabilities of the pretrained models and reduce the noise caused by irrelevant content.

Specifically, for the document $D$ containing sentences whose references are $(R_1, R_2, \cdots, R_m)$, we first obtain their corresponding context-free representations $(r_1, r_2, \cdots, r_m)$ through the encoder. We suppose the more relevant sentence in the context should have a higher similarity with the current sentence. And the similarity score between $R_i$ and $R_j$ is calculated as follows:

$$\text{Sim}(R_i, R_j) = \text{Cosine\_similarity}(r_i, r_j)\alpha^{|i-j|}$$

where $\alpha$ is a decay factor for the distance between two sentences. For the current sentence, most relevant sentences are selected in order according to the similarity above, until the number of sentences meets the requirement or the length is about to exceed 512. The selected contexts and the current sentence are then concatenated in the original order, followed by similar training, as shown in Figure 1.

## 3 Experiments

The following experiments and evaluations are based on the annotated dataset provided by WMT Metrics Shared Tasks [2] and MTME toolkit [3]. Our Cont-COMET metric has been trained upon the COMET-21 model proposed by Unbabel [4]. The detailed hyper-parameters and training settings are described in Appendix A.

### 3.1 Datasets

WMT Metrics Shared Tasks have significant impacts on MT evaluation and release large-scale annotated data each year, promoting state-of-the-art research. Although the integration of contextual information into the human annotation has been applied since 2019, the official dataset from WMT 2019 misses much meta information, and Direct Assessment (DA) scores in WMT 2021 and

---

[2]https://wmt-metrics-task.github.io.
[3]https://github.com/google-research/mt-metrics-eval.
[4]https://unbabel.github.io/COMET/html/index.html.

---

| Context Setting | 21 Sys | 21 Seg | 22 Sys | 22 Seg |
|---|---|---|---|---|
| COMET-21 | 0.657 | **0.282** | 0.910 | **0.349** |
| + 2 prev sentences | 0.656 | 0.240 | 0.940 | 0.283 |
| + 1 prev & 1 next sentences | 0.655 | 0.258 | 0.944 | 0.322 |
| + 4 prev sentences | 0.657 | 0.232 | 0.944 | 0.291 |
| + 2 prev & 2 next sentences | 0.656 | 0.237 | 0.950 | 0.301 |
| + 6 prev sentences | 0.656 | 0.222 | 0.946 | 0.283 |
| + 3 prev & 3 next sentences | **0.659** | 0.230 | 0.952 | 0.292 |
| + 8 prev sentences | 0.654 | 0.224 | 0.947 | 0.283 |
| + 4 prev & 4 next sentences | 0.658 | 0.229 | **0.953** | 0.292 |

Table 1: The results of COMET-21 under different settings of context usage during testing. Sys, Seg, 21, and 22 refer to the system-level and segment-level correlations, WMT21 and WMT22 test sets, respectively.

WMT 2022 were not available. Therefore, we collect the publicly released DA scores from WMT 2020, matching the document information, as our context-aware training set. Moreover, we use the Multidimensional Quality Metrics (MQM) data from WMT 2020 as our validation set, and MQM data from WMT 2021 and WMT 2022 for evaluating our trained metric, the same as previous work. More details are stated in Appendix B.

### 3.2 Evaluation Methods

To evaluate the performance of our metric, we measured system-level and segment-level consistency with the gold standard of human judgment. We use Kendall's Tau (Freitag et al., 2022) for the segment-level evaluation and Pearson correlation coefficient for the system-level evaluation, which align with the assessments in WMT Metrics Shared Tasks. The system-level pairwise accuracy is also additionally considered, and all the definitions are same as Freitag et al. (2021).

## 4 Results and Discussions

We argue the conclusions of Vernikos et al. (2022) that using two preceding sentences during inference yields the best performance are not adequately substantiated. Their work was based on the findings of previous studies (Kim et al., 2019; Castilho et al., 2020) and they did not attempt more sentences or other context settings. Therefore, we conducted similar but more in-depth experiments, taking into account the role of both preceding and subsequent context, as well as different numbers of sentences.

The results of the original COMET-21 model using additional context sentences during inference without extra training are presented in Table 1. For

| Model | WMT21.TED | | | WMT21.News | | | WMT22 | | |
|---|---|---|---|---|---|---|---|---|---|
| | Sys.Cor | Seg.Cor | Sys.Acc | Sys.Cor | Seg.Cor | Sys.Acc | Sys.Cor | Seg.Cor | Sys.Acc |
| COMET-21 | 0.653 | 0.254 | 76.92 | 0.661 | **0.311** | 70.85 | 0.910 | 0.349 | 83.94 |
| + 3 prev & 3 next sentences | 0.672 | 0.211 | 77.73 | 0.645 | 0.249 | 69.64 | **0.952** | 0.292 | **85.77** |
| Our Cont-COMET | | | | | | | | | |
| 2 context sentences selection | 0.665 | 0.265 | 78.54 | 0.664 | 0.306 | **71.66** | 0.938 | 0.353 | 84.31 |
| 4 context sentences selection | 0.669 | 0.265 | 78.54 | 0.663 | 0.306 | 70.45 | 0.941 | 0.353 | 84.67 |
| 6 context sentences selection | **0.678** | **0.265** | **78.95** | **0.665** | 0.303 | 71.26 | 0.943 | **0.354** | 84.31 |
| 8 context sentences selection | 0.677 | 0.263 | **78.95** | 0.664 | 0.303 | 70.85 | 0.943 | 0.354 | 84.67 |

Table 2: The results of our Cont-COMET metric with context selection of different numbers of sentences. Sys.Cor, Seg.Cor and Sys.Acc refer to the system-level correlation, segment-level correlation, and system-level pairwise accuracy according to human judgments, respectively.

brevity, the results of two domains, TED and News in WMT 2021 are averaged, which is the same for the following Table 3. Moreover, each test set actually involves three language pairs: en-de, en-ru, and zh-en. Due to space limitations, we average their results to represent the performance of the corresponding test set in our experiments. More details and complete results are shown in Appendix C with other baseline metrics.

The results demonstrate that the combined effect of preceding and subsequent context yields better performance when just using the same number of previous sentences. Furthermore, although this method without additional training improves system-level performance compared to the original COMET-21, as more contexts are used, it exhibits significant declines in correlations at the segment level. Table 2 shows the detailed results of our trained Cont-COMET metric. We also evaluate the different numbers of context sentences during the context selection method. Similar to the results in Table 1, Cont-COMET achieves the best performance when using context containing approximately six sentences. It may be attributed to the degraded performance of the encoder in processing longer content. Moreover, compared to previous settings without training in Table 1, Cont-COMET significantly improves the segment-level correlation. In terms of system-level performance, except for a slight drop on WMT22, there are also improvements on the other two test sets. Overall, our Cont-COMET demonstrates superior performance to the original COMET-21 in all aspects of evaluation.

## 5 Ablation Study

Furthermore, we conduct the ablation study to verify the effectiveness of different training ap-

| Different Training | 21 Sys | 21 Seg | 22 Sys | 22 Seg |
|---|---|---|---|---|
| Our Cont-COMET | 0.672 | 0.284 | 0.943 | 0.354 |
| w/o context selection | 0.669 | 0.284 | 0.942 | 0.353 |
| w/o context sentences | 0.665 | 0.230 | 0.953 | 0.296 |
| w/ previous sentences | 0.665 | 0.280 | 0.936 | 0.351 |
| w/ hypothesis context | 0.643 | 0.284 | 0.923 | 0.358 |

Table 3: The results of different training approaches for the ablation study, involving context with six sentences.

proaches, with corresponding results shown in Table 3. Our context selection method obtains slight improvements at both system level and segment level. Moreover, we similarly train and evaluate the metric only using the previous sentences. To prove the necessity of introducing context during training, we additionally train the contrast metric without context but use the same context as Cont-COMET in testing. We also attempt the concatenation of the hypothesis with its own context rather than that of the corresponding reference, which has been mentioned in Section 2.1. The results of the latter two approaches demonstrate significant drops in the segment-level and system-level correlations, respectively. When only utilizing previous sentences, it is inferior in all aspects of assessment, consistent with the results in Table 1. Overall, our proposed context-aware training has been proven effective and essential for the MT evaluation.

## 6 Conclusions

In this work, we explore context-aware training for MT automatic evaluation by incorporating context content into the encoding of sentences to be evaluated. We believe that our method can better align with the annotation settings of the existing large-scale annotated dataset, which was ignored in previous studies. Additionally, we introduce a context selection method to extract the most relevant

content from the document to enhance the contextual information, and train our Cont-COMET metric accordingly. The experiments and ablation studies prove the effectiveness of context utilization and the overall improvement of our metric. In the future, we hope our work will encourage the community to take notice of context-aware evaluation and conduct more in-depth research, and our proposed metric will be applied to document machine translation studies.

# 7  Limitations

The available annotated data for MT automatic evaluation that involves context information is insufficient for the time being. So our proposed metric may not be trained adequately, and efforts at related data augmentation can be attempted. Moreover, current neural metrics cannot really encode the entire document content, due to the length limitations of commonly used pretrained language models. In future research, we will explore more specific and novel methods to further take advantage of the context for MT evaluation.

# 8  Acknowledgements

This work was supported by National Key R&D Program of China (2021YFF0901502), National Science Foundation of China (No. 62161160339), State Key Laboratory of Media Convergence Production Technology and Systems and Key Laboratory of Science, Technology and Standard in Press Industry (Key Laboratory of Intelligent Press Media Technology). We appreciate the anonymous reviewers for their helpful comments, and everyone who has provided assistance in this work. Xiaojun Wan is the corresponding author.

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

## A   Training Settings

Since our context-aware training data only involves DA scores from WMT 2020, which is far less than those of the existing neural metrics, we conduct continuing training on the COMET-21 model (Rei et al., 2021). It has been trained on WMT15–20 DA scores, and we only fine-tune the parameters of the regression estimator to prevent overfitting. The encoder and embedding layer are frozen during training. The detailed hyper-parameters of our trained metric are shown in Table 4. During the concatenation of the context and current sentence, the separate tokens <SEP> are added. Moreover, when using the context selection method, we attempted different sentence representations to calculate similarity scores. In the original work, the final representations were obtained by combining the representations from different layers through layer-wise attention. In our experiments, we employ the average representations from the first and last layers, which yields better performance.

## B   Dataset Statistics

In fact, the officially released datasets from WMT Metrics Shared Tasks have multiple sources that are inconsistent with each other and miss some meta information. After the comparison of data quality, we finally selected data from the MTME toolkit, filtered out the samples without annotated scores, and

| Hyper-Parameters | |
| --- | --- |
| optimizer | AdamW |
| learning_rate | 5e-6 |
| dropout | 0.15 |
| batch_size | 16 |
| epoch | 1 |
| layer | mix |
| pool | avg |
| keep_embeddings_frozen | True |

Table 4: The hyper-parameters during training.

| Language Pair | #Assessment |
| --- | --- |
| Czech-English | 7968 |
| German-English | 10205 |
| Inuktitut-English | 35652 |
| Japanese-English | 9930 |
| Khmer-English | 18560 |
| Polish-English | 14014 |
| Pashto-English | 19033 |
| Russian-English | 11892 |
| Tamil-English | 13958 |
| Chinese-English | 34000 |
| English-Czech | 18434 |
| English-German | 24106 |
| English-Inuktitut | 35652 |
| English-Japanese | 12000 |
| English-Polish | 15000 |
| English-Russian | 20020 |
| English-Tamil | 16000 |
| English-Chinese | 19852 |

Table 5: The detailed statistics of our collected context-aware training dataset.

matched the corresponding contextual information. In general, the context-aware dataset we collected includes 18 language pairs, and the statistics are shown in Table 5. We will release the dataset for future in-depth research. On the other hand, the MQM data used for evaluating our trained metric also comes from MTME and involves three language pairs: en-de, en-ru, and zh-en. These data are annotated by professional evaluators with context content, and also serve as the official test datasets for the corresponding WMT Metrics Shared Task. And we follow the official evaluation framework with MTME.

## C    Additional Results

We add more detailed experimental results in Table 6, including the additional system-level pairwise accuracy. As For the COMET-21 and the different integration of context in the first seven lines of Table 6, they have not received additional training and have just been tested with the described content. On the other hand, we attempt several different training approaches, which are included in the part of our Cont-COMET in Table 6. They correspond to the experiments in Table 3 and involve different numbers of context sentences. Moreover, the results with different language pairs compared with other baseline metrics are displayed in Table 7 for a more comprehensive comparison.

| Model | WMT21.TED | | | WMT21.News | | | WMT22 | | |
|---|---|---|---|---|---|---|---|---|---|
| | Sys.Cor | Seg.Cor | Sys.Acc | Sys.Cor | Seg.Cor | Sys.Acc | Sys.Cor | Seg.Cor | Sys.Acc |
| COMET-21 | 0.653 | 0.254 | 76.92 | 0.661 | 0.311 | 70.85 | 0.910 | 0.349 | 83.94 |
| + 1 prev & 1 next sentences | 0.656 | 0.246 | 76.52 | 0.653 | 0.270 | 70.85 | 0.944 | 0.322 | 85.40 |
| + 2 prev sentences | 0.661 | 0.232 | 78.14 | 0.651 | 0.248 | 70.45 | 0.940 | 0.300 | 86.13 |
| + 2 prev & 2 next sentences | 0.665 | 0.231 | 77.73 | 0.648 | 0.242 | 68.83 | 0.950 | 0.301 | 85.77 |
| + 4 prev sentences | 0.670 | 0.207 | 77.73 | 0.645 | 0.257 | 70.04 | 0.944 | 0.291 | 86.13 |
| + 3 prev & 3 next sentences | 0.672 | 0.211 | 77.73 | 0.645 | 0.249 | 69.64 | 0.952 | 0.292 | 85.77 |
| + 6 prev sentences | 0.669 | 0.201 | 77.73 | 0.643 | 0.243 | 70.85 | 0.946 | 0.283 | 86.13 |
| Our Cont-COMET | | | | | | | | | |
| 2 context sentences selection | 0.665 | 0.265 | 78.54 | 0.664 | 0.306 | 71.66 | 0.938 | 0.353 | 84.31 |
| 1 prev & 1 next sentences | 0.659 | 0.264 | 77.73 | 0.661 | 0.306 | 71.66 | 0.938 | 0.353 | 84.67 |
| 2 prev sentences | 0.664 | 0.264 | 77.73 | 0.664 | 0.302 | 72.47 | 0.933 | 0.351 | 85.40 |
| w/o context training (2 sentences) | 0.660 | 0.247 | 77.73 | 0.665 | 0.271 | 70.85 | 0.945 | 0.324 | 84.67 |
| w/ hypothesis context (2 sentences) | 0.653 | 0.262 | 76.52 | 0.644 | 0.301 | 70.45 | 0.925 | 0.352 | 83.21 |
| 4 context sentences selection | 0.669 | 0.265 | 78.54 | 0.663 | 0.306 | 70.45 | 0.941 | 0.353 | 84.67 |
| 2 prev & 2 next sentences | 0.668 | 0.263 | 77.73 | 0.662 | 0.304 | 71.26 | 0.942 | 0.353 | 84.31 |
| 4 prev sentences | 0.674 | 0.264 | 77.73 | 0.660 | 0.302 | 71.26 | 0.935 | 0.351 | 85.04 |
| w/o context training (4 sentences) | 0.667 | 0.234 | 78.14 | 0.660 | 0.245 | 70.04 | 0.951 | 0.305 | 85.04 |
| w/ hypothesis context (4 sentences) | 0.656 | 0.266 | 77.73 | 0.637 | 0.299 | 70.85 | 0.923 | 0.357 | 83.94 |
| 6 context sentences selection | 0.678 | 0.265 | 78.95 | 0.665 | 0.303 | 71.26 | 0.943 | 0.354 | 84.31 |
| 3 prev & 3 next sentences | 0.675 | 0.263 | 78.54 | 0.662 | 0.306 | 71.26 | 0.942 | 0.353 | 84.31 |
| 6 prev sentences | 0.672 | 0.261 | 78.54 | 0.658 | 0.298 | 70.45 | 0.936 | 0.351 | 85.04 |
| w/o context training (6 sentences) | 0.675 | 0.214 | 78.14 | 0.656 | 0.246 | 69.64 | 0.953 | 0.296 | 85.40 |
| w/ hypothesis context (6 sentences) | 0.655 | 0.268 | 77.73 | 0.631 | 0.301 | 70.85 | 0.923 | 0.358 | 84.31 |

Table 6: The detailed results of our Cont-COMET metric with different training approaches, together with the context-aware testing of the COMET-21 model. Sys.Cor, Seg.Cor and Sys.Acc refer to the system-level correlation, segment-level correlation, and system-level pairwise accuracy according to human judgments, respectively.

| Model | System-level Correlation | | | Segment-level Correlation | | | System-level Accuracy |
|---|---|---|---|---|---|---|---|
| | En-De | En-Ru | Zh-En | En-De | En-Ru | Zh-En | |
| | | | WMT21.TED | | | | |
| COMET-21 | 0.781 | 0.872 | 0.306 | 0.268 | 0.274 | 0.220 | 76.92 |
| + 3 prev & 3 next sentences | 0.783 | 0.897 | 0.336 | 0.202 | 0.268 | 0.163 | 77.73 |
| Our Cont-COMET | 0.795 | 0.889 | 0.350 | 0.263 | 0.289 | 0.243 | 78.95 |
| BLEU (Papineni et al., 2002) | 0.620 | 0.828 | 0.324 | 0.113 | 0.112 | 0.092 | 74.1 |
| chrF (Popovic, 2015) | 0.471 | 0.825 | 0.363 | 0.146 | 0.189 | 0.124 | 71.3 |
| YiSi-1 (Lo, 2019) | 0.414 | 0.905 | 0.310 | 0.212 | 0.204 | 0.195 | 75.7 |
| BERTScore (Zhang et al., 2020) | 0.506 | 0.831 | 0.306 | 0.189 | 0.185 | 0.199 | 72.1 |
| BLEURT-20 (Sellam et al., 2020) | 0.739 | 0.868 | 0.239 | 0.283 | 0.255 | 0.224 | 74.9 |
| | | | WMT21.News | | | | |
| COMET-21 | 0.812 | 0.654 | 0.517 | 0.253 | 0.307 | 0.372 | 70.85 |
| + 3 prev & 3 next sentences | 0.832 | 0.636 | 0.467 | 0.197 | 0.276 | 0.273 | 69.64 |
| Our Cont-COMET | 0.823 | 0.676 | 0.495 | 0.247 | 0.305 | 0.356 | 71.26 |
| BLEU (Papineni et al., 2002) | 0.937 | 0.507 | 0.310 | 0.083 | 0.120 | 0.176 | 74.1 |
| chrF (Popovic, 2015) | 0.846 | 0.783 | 0.302 | 0.114 | 0.193 | 0.201 | 74.5 |
| YiSi-1 (Lo, 2019) | 0.789 | 0.761 | 0.515 | 0.172 | 0.233 | 0.302 | 73.7 |
| BERTScore (Zhang et al., 2020) | 0.930 | 0.629 | 0.542 | 0.169 | 0.185 | 0.296 | 74.5 |
| BLEURT-20 (Sellam et al., 2020) | 0.802 | 0.768 | 0.563 | 0.264 | 0.286 | 0.354 | 75.3 |
| | | | WMT22 | | | | |
| COMET-21 | 0.856 | 0.917 | 0.958 | 0.333 | 0.362 | 0.351 | 83.94 |
| + 3 prev & 3 next sentences | 0.929 | 0.956 | 0.972 | 0.280 | 0.326 | 0.271 | 85.77 |
| Our Cont-COMET | 0.915 | 0.943 | 0.971 | 0.341 | 0.372 | 0.348 | 84.31 |
| BLEU (Papineni et al., 2002) | 0.179 | 0.724 | 0.594 | 0.169 | 0.140 | 0.145 | 70.8 |
| chrF (Popovic, 2015) | 0.346 | 0.815 | 0.630 | 0.214 | 0.168 | 0.147 | 73.4 |
| YiSi-1 (Lo, 2019) | 0.626 | 0.881 | 0.935 | 0.235 | 0.227 | 0.296 | 79.2 |
| BERTScore (Zhang et al., 2020) | 0.428 | 0.811 | 0.924 | 0.232 | 0.192 | 0.316 | 77.4 |
| BLEURT-20 (Sellam et al., 2020) | 0.719 | 0.959 | 0.938 | 0.344 | 0.359 | 0.361 | 84.7 |

Table 7: The additional results with different language pairs and other baseline metrics. And our Cont-COMET model shown here utilizes six context sentences selection.