# OpenReview forum: "Exploring Context-Aware Evaluation Metrics for Machine Translation"
_EMNLP/2023/Conference — EMNLP 2023 Findings_

### Official Review · Reviewer_7yHH · 2023-08-05

**Soundness:** 3

**Excitement:**

2: Mediocre: This paper makes marginal contributions (vs non-contemporaneous work), so I would rather not see it in the conference.

**Missing References:**

BlonDe: An Automatic Evaluation Metric for Document-level Machine Translation (Jiang et al., NAACL 2022)

**Paper Topic And Main Contributions:**

This work introduces a context-aware machine translation metric, Cont-COMET, built on top of the trained COMET metric used for machine translation evaluation, which has been shown to have better correlations with human judgments compared to string-based matching metrics such as BLEU. In this work, the authors experiment with including previous and next context sentences for both source and target reference sentences, using reference sentences as context for the machine-generated hypothesis translation. All of these are encoded using XLM-R and trained to produce a score, in the same way as COMET. Experiments are conducted using cosine similarity to select relevant sentences from the context since XLM-R can only process up to 512 tokens. Experiments demonstrate some improvements in scores compared to COMET.

**Questions For The Authors:**

- What kind of sentences were selected as context when cosine similarity was used? Was any manual analysis conducted to check the usefulness of the context?

**Reasons To Accept:**

As motivated by the paper, more work needs to be done towards having reliable context-aware metrics. So far there have been few model-based trained metrics that have incorporated context, so this work is a welcome step in that direction.

**Reasons To Reject:**

- The improvements demonstrated are marginal, and their advantage over COMET is unclear. COMET still outperforms their proposed metric in several cases. Statistical significance tests are not conducted, so it is unclear if the improvements are significant as claimed.
- Another claim that does not seem to have sufficient backing is that using a similar number of sentences for preceding and subsequent context performs the best. This conclusion seems to be based on a couple of improved results in Table 1, one of which is marginal, while all the rest of the results seem to show the contrary.
- One of the assumptions made for context selection is that sentences similar to the current sentence (based on cosine similarity) are more relevant context. There is a good chance this is unlikely, since useful context often contains information that is not present in the current sentence (e.g. antecedents of pronouns) and is therefore less likely to be similar.
- There is no language-wise breakdown of results in the paper. It is likely that the usefulness of context differs depending on the source and target languages and it seems to be a fairly big oversight to have a discussion of this missing.

**Reproducibility:**

3: Could reproduce the results with some difficulty. The settings of parameters are underspecified or subjectively determined; the training/evaluation data are not widely available.

**Reviewer Confidence:**

4: Quite sure. I tried to check the important points carefully. It's unlikely, though conceivable, that I missed something that should affect my ratings.

**Typos Grammar Style And Presentation Improvements:**

- It is claimed in the paper that WMT introduced context information for human judgments in 2019. This may need to be double-checked, since the document-level MT task was introduced in 2019; annotators were able to see context during evaluation much earlier.
- It is not clear that the results in Table 1 are for untrained COMET with additional context sentences. This needs to be mentioned more explicitly before presenting the results.
- Please also mention the language pairs in the test data in the main paper and perform statistical significance tests. Having a language-pair-wise breakdown will also help analysis.
- ll 206-208 mention that Vernikos et al (2022) do not provide comprehensive conclusions. It would be helpful to include what conclusions they do provide, and why this paper is more comprehensive in comparison.
- ll 029 were early proposed -> were earlier proposed
- 043-044 while the related document content was often involved in practice -> in which practice? Unclear what this sentence means.

---

> ### Author Rebuttal · Authors · 2023-08-29
>
> Thank you for the time and suggestions, and we will answer your questions as follows:
>
> > 1. The improvements demonstrated are marginal, and their advantage over COMET is unclear. COMET still outperforms their proposed metric in several cases. Statistical significance tests are not conducted, so it is unclear if the improvements are significant as claimed.
>
> The results reported in our paper were obtained by averaging results from multiple experiments with different random seeds, ensuring reliability. Therefore, we think the improvements from our approach can be identified as significant. Moreover, COMET only exhibits better performance in terms of segment-level correlation in the domain of news in WMT21, compared to the metrics we proposed.
>
> > 2. Another claim that does not seem to have sufficient backing is that using a similar number of sentences for preceding and subsequent context performs the best. This conclusion seems to be based on a couple of improved results in Table 1, one of which is marginal, while all the rest of the results seem to show the contrary.
>
> We agree with your opinion, and conducting experiments with different numbers of preceding and subsequent context sentences can enhance the clarity of our method. We will add more detailed context utilization settings in the appendix, including different combinations of numbers of context sentences.
>
> > 3. One of the assumptions made for context selection is that sentences similar to the current sentence (based on cosine similarity) are more relevant context. There is a good chance this is unlikely, since useful context often contains information that is not present in the current sentence (e.g. antecedents of pronouns) and is therefore less likely to be similar.
>
> We acknowledge that this idea is somewhat subjective, but our method was based on semantic similarity, which is intuitive and reasonable. It is because, during document-level translation, semantic considerations play an important role in handling context content. And our experiment results also indicate that our context selection method achieves certain improvements.
>
> > 4. There is no language-wise breakdown of results in the paper. It is likely that the usefulness of context differs depending on the source and target languages and it seems to be a fairly big oversight to have a discussion of this missing.
>
> We apologize for our oversight. Our initial experiment results actually contain the performance across different languages. However, due to space limitations in the short paper, we did not include these detailed results and relevant discussions. We will supplement them in the appendix.
>
> > 5. What kind of sentences were selected as context when cosine similarity was used? Was any manual analysis conducted to check the usefulness of the context?
>
> We used sentences from the reference to maintain the same setting as our context extension process. And we did not conduct additional manual analysis, which could be considered for further evaluation.
>
> > 6. Missing References and Typos Grammar Style And Presentation Improvements.
>
> Thank you very much for your reviews and the valuable suggestions that have helped us enhance the quality of our work. We will carefully address and improve upon these issues.

---

### Official Review · Reviewer_KJ38 · 2023-08-05

**Soundness:** 4

**Excitement:**

3: Ambivalent: It has merits (e.g., it reports state-of-the-art results, the idea is nice), but there are key weaknesses (e.g., it describes incremental work), and it can significantly benefit from another round of revision. However, I won't object to accepting it if my co-reviewers champion it.

**Paper Topic And Main Contributions:**

The paper proposes Cont-COMET a modification of the popular machine translation (MT) evaluation method COMET by adding consideration for previous and following sentence context during evaluation, as well as a context selection method that enables finding the most relevant context. Authors experiment with data from the WMT metrics shared tasks and report improvements in system-level and segment-level evaluations.

**Reasons To Accept:**

Substantial work has been done in an area where there's still room for improvement, promising results have been reported, at least on the system level compared to the original COMET-21 metric.

**Reasons To Reject:**

Would be interesting to see the proposed metric somehow compared with some other context-sensitive evaluation methods or even participants of the WMT metrics task.

**Reproducibility:**

3: Could reproduce the results with some difficulty. The settings of parameters are underspecified or subjectively determined; the training/evaluation data are not widely available.

**Reviewer Confidence:**

4: Quite sure. I tried to check the important points carefully. It's unlikely, though conceivable, that I missed something that should affect my ratings.

---

> ### Author Rebuttal · Authors · 2023-08-29
>
> Thank you for the time and suggestions, and we will answer your questions as follows:
>
> > 1. Would be interesting to see the proposed metric somehow compared with some other context-sensitive evaluation methods or even participants of the WMT metrics task.
>
> The existing context-sensitive evaluation methods are very limited. We only found the work of Doc-COMET, which has been introduced in our paper and mainly compared with our methods. Moreover, participants in the WMT metrics task almost pay attention to the evaluation of individual sentences, which includes certain limitations. We agree that adding more automatic evaluation metrics can demonstrate the improvements of our method. We will collect the relevant results and add a table in the appendix to present them.

---

### Official Review · Reviewer_844M · 2023-08-09

**Soundness:** 3

**Excitement:**

4: Strong: This paper deepens the understanding of some phenomenon or lowers the barriers to an existing research direction.

**Paper Topic And Main Contributions:**

The paper proposes a modification of the COMET MT evaluation metrics which adds context of the scored sentence into consideration. The paper improves the previous methods going in the similar direction by selection of the context sentences based not on sentence order but rather on sentence similarity. This is the main contribution of the paper.

**Reasons To Accept:**

The context sentence selection method is an interesting novel idea that can motivate other researchers to further extend it.
Blindly including previous and next sentences improves system level correlation but hurts segment level one. But the proposed method improves both.

**Reasons To Reject:**

The results are marginal, test of statistical significance could confirm whether the improvements are significant.

There are a few unexplored points of the presented method:
1. The sentence similarity is based on the reference. What happen if the selection is based on the hypothesis (or even the source)?
2. Inclusion of the  context hypothesis sentences basically means that the quality of their translation has some impact on the score of the hypothesis itself. This could (possibly) help system level correlation but can hurt the segment level one.
3. The number of context sentences is fixed. Another approach would be to use a similarity threshold and variable number of sentences.

**Reproducibility:**

4: Could mostly reproduce the results, but there may be some variation because of sample variance or minor variations in their interpretation of the protocol or method.

**Reviewer Confidence:**

3: Pretty sure, but there's a chance I missed something. Although I have a good feel for this area in general, I did not carefully check the paper's details, e.g., the math, experimental design, or novelty.

---

> ### Author Rebuttal · Authors · 2023-08-29
>
> Thank you for the time and suggestions, and we will answer your questions as follows:
>
> > 1. The results are marginal, test of statistical significance could confirm whether the improvements are significant.
>
> The results reported in our paper were obtained by averaging results from multiple experiments with different random seeds, ensuring reliability. Therefore, the improvements from our approach can be identified as significant.
>
> > 2. The sentence similarity is based on the reference. What happen if the selection is based on the hypothesis (or even the source)?
>
> Indeed, we did conduct these experiments previously, where we used sources or hypotheses as the objects for calculating similarity. However, the experiment's results were not satisfactory. We suppose that it might be attributed to our context extension mainly relying on references. Hence, maintaining the same context setting could yield the best improvement.
>
> > 3. Inclusion of the context hypothesis sentences basically means that the quality of their translation has some impact on the score of the hypothesis itself. This could (possibly) help system level correlation but can hurt the segment level one.
>
> We have already experimented with the context of hypothesis sentences in the ablation studies in Table 3 and displayed the corresponding results. The system-level correlation exhibited a noticeable decline, whereas the segment-level correlation did not. This point is interesting, and we intend to further explore it in the future.
>
> > 4. The number of context sentences is fixed. Another approach would be to use a similarity threshold and variable number of sentences.
>
> We agree with your opinion, which provides another approach worth exploring. We will supplement the results of relevant experiments for better comparison.

---

### Official Review · Reviewer_C6zw · 2023-08-12

**Soundness:** 3

**Excitement:**

3: Ambivalent: It has merits (e.g., it reports state-of-the-art results, the idea is nice), but there are key weaknesses (e.g., it describes incremental work), and it can significantly benefit from another round of revision. However, I won't object to accepting it if my co-reviewers champion it.

**Paper Topic And Main Contributions:**

This paper focuses on automatic evaluation metrics for machine translation, and proposes a context-aware automatic evaluation metric called Cont-COMET. This metric builds on the COMET framework and introduces additional contextual information, enhancing the utilization of information through a context selection mechanism. Finally, the paper validates the effectiveness of the metric through various experiments and analyzes the impact of different settings through ablation experiments.

**Reasons To Accept:**

1. The authors analyzed the impact of contextual information on machine translation evaluation and verified the hypotheses through reasonable experiments, providing a better automated evaluation metric.

2. The authors added the context selection module based on the COMET framework, which enables more comprehensive utilization of information within the allowed length of pretrained language model encoders.

3. The authors collected and collated a series of publicly available corpora, resulting in a high-quality corpus that can be used for automatic metric training.

**Reasons To Reject:**

1. The method proposed in the paper is slightly lacking in innovation, and perhaps further exploration of more efficient context utilization and modeling methods can be made.

2. Based on the results of the ablation experiments in the paper, the effect change brought by context selection is small, which is not consistent with the hypothesis in the previous text. The authors may add some additional explanations.

3. According to the results in Table 2 and Table 3, the improvements brought by the proposed approach compared to the COMET-21 metric are mostly due to the introduction of contextual information in training. The overall level of innovation is slightly insufficient.

4. The paper could consider adding some other automated evaluation metric baselines to highlight the improvement brought by the current approach.

5. It would be preferable to provide a brief introduction to the COMET framework in the paper.

**Reproducibility:**

4: Could mostly reproduce the results, but there may be some variation because of sample variance or minor variations in their interpretation of the protocol or method.

**Reviewer Confidence:**

4: Quite sure. I tried to check the important points carefully. It's unlikely, though conceivable, that I missed something that should affect my ratings.

---

> ### Author Rebuttal · Authors · 2023-08-29
>
> Thank you for the time and suggestions, and we will answer your questions as follows:
>
> > 1. The method proposed in the paper is slightly lacking in innovation, and perhaps further exploration of more efficient context utilization and modeling methods can be made.
>
> The main contribution of our work lies in observing the context issues in existing translation evaluation metrics research and proposing corresponding improved metrics and datasets. We hope we can help the community promote the relevant research more effectively. And our method may slightly lack innovation, so we submitted a short paper, leaving further exploration of more efficient context utilization and modeling methods for future research.
>
> > 2. Based on the results of the ablation experiments in the paper, the effect change brought by context selection is small, which is not consistent with the hypothesis in the previous text. The authors may add some additional explanations.
>
> In fact, as the context length used by the model increases, the performance gains from context-aware methods do diminish. We will supplement more relevant experimental results to demonstrate the effectiveness of our context selection method.
>
> > 3. According to the results in Table 2 and Table 3, the improvements brought by the proposed approach compared to the COMET-21 metric are mostly due to the introduction of contextual information in training. The overall level of innovation is slightly insufficient.
>
> The main contribution of our work lies in observing the context issues in existing translation evaluation metrics research and proposing corresponding improved metrics and datasets. We hope we can help the community promote the relevant research more effectively. The integration of contextual content into the automatic translation evaluation metric plays the most important role, and the context selection method is supplementary.
>
> > 4. The paper could consider adding some other automated evaluation metric baselines to highlight the improvement brought by the current approach.
>
> We agree that adding more automatic evaluation metric baselines can demonstrate the improvements of our method. We will collect the relevant results and add a table in the appendix to present them.
>
> > 5. It would be preferable to provide a brief introduction to the COMET framework in the paper.
>
> Given the limited space of the short paper, we only provided a very brief description of the COMET framework in the final submitted paper. And we will provide more detailed information in the appendix.

---

### Meta-Review · Area_Chair_ruWs · 2023-09-18

**Recommendation:** 3

**Metareview:**

Four reviewers gave the Soundness scores 3/3/4/3 and Excitement Scores 3/4/3/2. The authors have responded to all four reviewers and all reviewers acknowledged the author's responses.
All reviewers tend to agree that this paper has positive Soundness in addressing context-aware MT Evaluation, with a minimum score 3 - good level. However, most reviewers do not seem to be excited for this paper to appear in the main conference. It shall appear in the findings if possible.

---

### Decision · Program_Chairs · 2023-10-07

**Decision:**

Accept-Findings

**Comment:**

Four reviewers gave the Soundness scores 3/3/4/3 and Excitement Scores 3/4/3/2. The authors have responded to all four reviewers and all reviewers acknowledged the author's responses.
All reviewers tend to agree that this paper has positive Soundness in addressing context-aware MT Evaluation, with a minimum score 3 - good level. However, most reviewers do not seem to be excited for this paper to appear in the main conference. It shall appear in the findings if possible.